# Evaluating Routine Blood Tests According to Clinical Symptoms and Diagnostic Criteria in Individuals with Myalgic Encephalomyelitis/Chronic Fatigue Syndrome

**DOI:** 10.3390/jcm10143105

**Published:** 2021-07-14

**Authors:** Ingrid H. Baklund, Toril Dammen, Torbjørn Åge Moum, Wenche Kristiansen, Daysi Sosa Duarte, Jesus Castro-Marrero, Ingrid Bergliot Helland, Elin Bolle Strand

**Affiliations:** 1Department of Behavioural Medicine, Faculty of Medicine, University of Oslo, 0315 Oslo, Norway; i.h.m.e.baklund@studmed.uio.no (I.H.B.); Toril.dammen@medisin.uio.no (T.D.); t.a.moum@medisin.uio.no (T.Å.M.); 2CFS/ME Center, Division of Medicine, Oslo University Hospital, 0318 Oslo, Norway; wenckr@ous-hf.no (W.K.); daysos@ous-hf.no (D.S.D.); 3CFS/ME Unit, Division of Rheumatology, Vall d’Hebron Research Institute, Universitat Autònoma de Barcelona, 08035 Barcelona, Spain; jesus.castro@vhir.org; 4National Advisory Unit for CFS/ME, Rikshospitalet, Oslo University Hospital, Rikshospitalet OUS, 0372 Oslo, Norway; ihelland@ous-hf.no; 5Faculty of Health, VID Specialized University, 0370 Oslo, Norway

**Keywords:** myalgic encephalomyelitis/chronic fatigue syndrome, routine blood tests, diagnostic criteria, functional status, creatinine, creatine kinase

## Abstract

There is a lack of research regarding blood tests within individuals with Myalgic Encephalomyelitis/Chronic Fatigue Syndrome (ME/CFS) and between patients and healthy controls. We aimed to compare results of routine blood tests between patients and healthy controls. Data from 149 patients diagnosed with ME/CFS based on clinical and psychiatric evaluation as well as on the DePaul Symptom Questionnaire, and data from 264 healthy controls recruited from blood donors were compared. One-way ANCOVA was conducted to examine differences between ME/CFS patients and healthy controls, adjusting for age and gender. Patients had higher sedimentation rate (mean difference: 1.38, 95% CI: 0.045 to 2.714), leukocytes (mean difference: 0.59, 95% CI: 0.248 to 0.932), lymphocytes (mean difference: 0.27, 95% CI: 0.145 to 0.395), neutrophils (mean difference: 0.34, 95% CI: 0.0 89 to 0.591), monocytes (mean difference: 0.34, 95% CI: 0.309 to 0.371), ferritin (mean difference: 28.13, 95% CI: −1.41 to 57.672), vitamin B12 (mean difference: 83.43, 95% CI: 62.89 to 124.211), calcium (mean difference: 0.02, 95% CI: −0.02 to 0.06), alanine transaminase (mean difference: 3.30, 95% CI: −1.37 to -7.971), low-density lipoproteins (mean difference: 0.45, 95% CI: 0.104 to 0.796), and total proteins (mean difference: 1.53, 95% CI: −0.945 to 4.005) than control subjects. The patients had lower potassium levels (mean difference: 0.11, 95% CI: 0.056 to 0.164), creatinine (mean difference: 2.60, 95% CI: 0.126 to 5.074) and creatine kinase (CK) (mean difference: 37.57, 95% CI: −0.282 to 75.422) compared to the healthy controls. Lower CK and creatinine levels may suggest muscle damage and metabolic abnormalities in ME/CFS patients.

## 1. Introduction

Myalgic encephalomyelitis, also known as chronic fatigue syndrome (ME/CFS) is a debilitating disease and common symptoms are post-exertional malaise (PEM], headaches, muscle- and joint pain, dyspnoea, nausea, and flu-like symptoms [1,2]. It is affecting all social and racial/ethnic groups, although possibly women more frequently than men [3,4]. The severity of the illness ranges, from ambulant to housebound [5]. A 2020 EUROMENE review found that prevalence ranged from 0.1–2.2% [6]. An American report from 2015, summarizing more than 9000 papers about illness, concluded that “ME/CFS is a serious, chronic, complex, and multisystem disease that frequently and dramatically limits the daily activities of affected patients” [7]. Twenty-five percent of patients become house- or bedbound at some point of their illness course [8].

The illness burden also involves a great personal and societal economic loss. ME/CFS is estimated to affect over 2.5 million across Europe. The condition often results in diminished functionality and increased economic impact. Despite high prevalence rates and disabling nature of the illness, few studies have examined the economic impact at the individual level and the societal cost across Europe [9].

Despite the severe nature of the disease, the pathophysiology is still largely unknown. No cure or specific treatment exists, nor are there any specific biomarkers [10,11].

Most studies on different haematological and biochemical tests reveal that in most cases, no difference is found between ME/CFS patients and healthy controls [11,12,13,14,15,16,17,18,19,20,21,22,23,24,25,26,27,28,29,30]. However, one study reported reduced creatine kinase (CK) levels and a higher sedimentation rate (SR) and thrombocytes in patients compared to normal controls [11]. Furthermore, an elevated neutrophil count has been reported [13,31], and elevated white blood cells [32], monocytes [32], ferritin [15], triglycerides [18,31], mean corpuscular hemoglobin concentration (MCV) [12], albumin [31], C-reactive protein (CRP) [11,33,34], thyroid stimulating hormone (TSH) [21,31], alkaline phosphatase (ALP) [32], antinuclear Antibodies IIF [32], and the complement factors C3 and C4 [35] have also been found in patients compared to controls. A reduced glucose [36], phosphate [31], iron and transferrin saturation [14], vitamin B9 [37], high-density lipoprotein-cholesterol (HDL-cholesterol) [15,18,31], and lower cortisol have also been reported [38,39,40,41,42,43]. Free T4 has been found to be both elevated [44] and reduced [15], which is also the case for immunoglobulin G (IgG subclass 1 and 2) [24,26,30,32,45,46]. IgG subclasses 3 and 4, however, were reduced [24,45,46,47], while Immunoglobulin M (IgM) and Immunoglobulin A (IgA) were found to be elevated [32]. Some of these studies have included routine blood tests, or the equivalent [11,13,17,18,32], with various routine test panels, whereas other studies have investigated more specific hypotheses with a limited number of specific blood tests tailored to the hypothesis or in order to exclude fatigue-related conditions.

Several different diagnostic criteria are in clinical use, ranging from the strict International Consensus Criteria (ICC) that capture a condition with more severe symptoms than the other criteria [1], the more lenient Canadian Consensus Criteria (CCC) [3], and the most liberal Fukuda definition [2]. Most patients that fulfil the CCC will also fulfil the Fukuda definition, but the patients fulfilling the CCC may have a higher frequency and severity of functional impairment and physical and cognitive symptoms than those fulfilling the Fukuda criteria [7]. The CCC, and in particular ICC, claim to achieve a more narrow selection of patients, conforming to a hypothesis-specific pathophysiology [48]. It is considered likely that all ME/CFS case definitions capture conditions with different or multifactorial pathogenesis [48], but to the best of our knowledge there is no previous study that has explored differences in blood tests according to various diagnostic criteria for ME/CFS.

Although guidelines for blood sample tests in the diagnostic assessments of ME/CFS do exist, research is sparse regarding how the results of these tests differ from those of healthy controls and across patient characteristics. To address this knowledge gap, we aim to (1) compare the results of routine blood samples from patients and healthy controls (blood volunteer donors); (2) explore the correlation between the blood tests results, illness severity and duration within the patient group; and (3) compare results of routine blood tests between those who fulfill the ICC [1] vs. those only fulfilling the CCC [3] and/or the Fukuda case definition [2].

## 2. Materials and Methods

### 2.1. Participants

In this prospective observational cohort study, a total of 149 ME/CFS patients were consecutively referred to a tertiary care center for evaluation (Oslo University Hospital, Aker, Norway), and 264 healthy volunteer donors, between March 2013 and June 2019, were asked to participate in a thematic register and Biobank for research purposes. Patients had to fulfill the Canadian Consensus criteria (CCC) [3] as applied by a clinician, as well as the following inclusion criteria: aged 18–65 years old and able to understand and speak the Norwegian language. They were evaluated for eligibility and asked to participate during their second consultation by a physician.

From March 2013 to August 2015, 34 patients were included in the study. Unfortunately, relevant data for estimating a participation rate was not collected during this period. From August 2015 to 2019, a total of 288 patients were evaluated for study participation. Two-hundred-and thirteen were considered eligible for study participation at this time. One hundred and seventy-one (80%) agreed to participate, while 42 (20%) declined the request for participation. None were excluded because of age or language, but eight patients were excluded from the current dataset because they did not fulfill any ME/CFS criteria according to their DSQ responses. Of the 171 patients that consented to participation, 48 (28%) did not show up for further assessment and were thus excluded from the study. Most of them reported orally that they felt too ill or fatigued to attend. Thus, 115 patients were included during this period with an estimated participation rate of 40% (115/288). The number of included patients were 149 for the whole period from 2013–2019.

The patients gave a blood sample to the ME/CFS research Biobank and filled out questionnaires with information for the ME/CFS thematic research register. This included clinical and demographic information on patient history and treatment, epidemiology, work/social status, and occupation and DSQ.

The healthy control group consisted of 264 first-time blood volunteer donors at the blood bank at Oslo University Hospital. They were evaluated and no sign of any medical illness was found. They filled out similar questionnaires as the patients and were recruited and assessed during the same time period as the inclusion period for the ME/CFS patients. Both patients and controls were recruited from the area: south-eastern Norway (Helse Sør-Øst). 

### 2.2. ME/CFS Assessment and Diagnosis

The Canadian Consensus Criteria (CCC) [3] as applied by clinicians were used as inclusion criteria. This was assessed during a clinical interview by physicians highly experienced in ME/CFS diagnostics and all patients obtained their diagnosis after a thorough evaluation in interdisciplinary expert groups. In order to exclude somatic and/or psychiatric conditions that could explain the symptoms, several blood tests were taken, and a clinical psychological interview covering diagnostic assessment was carried out.

### 2.3. Measures

The DePaul Symptom Questionnaire (DSQ) is a 99 items self-report symptom questionnaire originally developed in order to meet the need for more reliable diagnostic categorization of ME/CFS for research purposes [49]. The DSQ can classify patients according to three diagnostic criteria sets: the Fukuda, the CCC and the ICC criteria. For participation in the current study the patients had to fulfil at least one of the diagnostic criteria according to the DSQ, in addition to the Canadian Consensus criteria (CCC) used in the clinical interview. The DSQ assesses information on frequency, severity, onset and duration of symptoms and contains questions on self-reported functioning level classified as very severely or severely impaired, as moderately or as mild degree of impairment. The DSQ is developed from a CFS questionnaire with good inter-rater and test–retest reliability and able to distinguish between CFS, Major Depressive Disorders, and healthy controls [50]. The DSQ has acceptable convergent and discriminant validity [51], test–retest reliability [52], sound psychometric properties to correctly classify ME/CFS within the CCC [53], excellent internal reliability and is able to differentiate between patients and controls [54]. It was translated into Norwegian and retranslated by a professional translator, with permission from the developer (Prof. Leonard A. Jason, DePaul University, Chicago, IL, USA), and reviewed by researchers and pre-tested in smaller groups of patients [53]. This version has been found useful for detecting and screening symptoms consistent with a CCC diagnosis showing a sensitivity of 98% and a specificity of 38% [53].

The questionnaires were completed by pen and paper at home before being delivered at the hospital at the appointment for blood sampling. To prevent missing data, a research nurse reviewed the questionnaires and the patients were requested to fill in missing data. Data were collected from the self-report questionnaire DSQ and blood tests. Questions from DSQ were applied for categorization of the patient groups that were included in the statistical analyses.

### 2.4. Patient Groupings: Functional Status, Illness Duration, and Diagnostic Criteria

Function status: The DSQ question 79 was used to categorize patients according to function impairment level (severe, moderate, and mild). The function level categorized as ‘severe’ was defined as responding positively to either “I am not able to work or do anything, and I am bedridden” or “I can walk around the house, but I cannot do light housework” whereas ‘moderate’ was defined as responding positively to “I can do light housework, but I cannot work part-time”. The final three statements, “I can only work part-time at work or on some family responsibilities”, “I can work full time and finish some family responsibilities but I have no energy left for anything else”, and “I can do all work or family responsibilities without any problems with my energy” were all categorized as ‘mild’. The ‘severe’ category comprised 43 patients (28.9%), the ‘moderate’ 75 patients (50.3%), while 31 patients (20.8%) were included in the ‘mild’ category.

Illness duration: Illness duration was assessed by the DSQ question 69 (“How long ago did your problem with fatigue/energy begin?”) and categorized according to the following responses; 1–2 years (9.4%), >2 years (79.2%) and problems starting in childhood (11.4%).

Case criteria applied: The CCC were used as inclusion criteria when the patients initially were diagnosed by the clinicians. In the current study DSQ was applied for diagnostic classification based on different diagnostic criteria and this revealed that all the 149 patients fulfilled the Fukuda case definition, and 93.3% (*n* = 139) fulfilled the CCC, and 63.1% (*n* = 94) patients fulfilled the ICC criteria.

Case criteria groupings: Patients were divided into one of two following case criteria groups: “Non-ICC” that comprised those 55 patients (36.9%) who did not fulfill ICC, but only Fukuda and CCC whereas the “ICC” group consisted of the 94 patients (63.1%) fulfilling all three criteria included the ICC.

### 2.5. Blood Collection and Processing

Blood samples were collected by experienced nurse at the ME/CFS outpatient clinic in Oslo, or bedside for the most severely ill. The samples were delivered to the central laboratory at Aker Hospital within 30 min from collection and analyzed consecutively. Some serum samples were transported refrigerated to other hospitals for further processing. Standard OUS laboratory protocols were used for all collection, tests, and transport.

### 2.6. Ethics

All participants were informed about the purposes of the study and they signed a written informed consent form. The study and all data collection, including Biobank sampling and thematic register were approved by the Institutional Review Board at Oslo University Hospital (ref: 2011/8355) and the local Regional Committee for Medical and Health Research Ethics (REC) (REC 2011/473, and REC South-East, 2017/375).

### 2.7. Statistical Analysis

SPSS (SPSS Inc. Released 2009, PASW Statistics for Windows, Version 25.0. SPSS Inc., Chicago, IL, USA) was used for all statistical analyses. Descriptive statistics were conducted for demographics—i.e., age, gender, body mass index (BMI) and level of education. Analyses of covariance (ANCOVAs) were performed to estimate controlled mean differences between patients and controls for the various biological variables (treated as dependent variables). Gender and age showed significant associations with the patient-control dichotomy as well as with duration and level of functioning among patients and thus were routinely controlled for in the ANCOVAs (procedure UNIANOVA in SPSS) and in linear regressions.

Data plots were inspected for outliers and outliers or extreme values and—when present—these were routinely removed from all dependent variables, in concrete terms by eliminating scores more than three standard deviations above or below the overall mean. Such a trimming of extreme values on average reduced the number of subjects with valid scores by less than 1%. In addition, closer inspection revealed that *p*-values for the patient versus controls tests were hardly affected at all by the trimming of extreme scores. Mean differences between groups in continuous variables when tested by *t*-tests (as in the present paper) typically require normally distributed variables within groups, but there is considerable robustness to deviations from normality overall. Log-transformed versions of dependents were visually inspected and yielded almost identical *p*-values (for *t*-tests) and within-group means (also when adjusted by ANCOVAs) are reported using the untransformed metric. Units of measurement for the dependent variables are shown as means and standard deviations in all tables. For illustrative purposes confidence intervals (CI) have been added after controlled means for patients and controls in Table 2.

Effect sizes for dichotomous and trichotomous independents in Tables 2 and 3 are cited as “eta”, i.e., the square of root of the variance explained by the groups comprising an independent, controlling for possible covariates. Effect sizes for linear trends in ordered trichotomous independents (function status, illness duration), are cited as standardized betas obtained by OLS. Levels of significance for effect sizes are cited as exact *p*-values (Table 2) and routinely categorized (Table 3).

## 3. Results

### 3.1. Study Population Characteristics

Demographic and clinical characteristics of the patients and controls are presented in Table 1. A significant difference in age (*p* < 0.001) and gender (*p* = 0.02) between patients and controls were found and therefore corrected for in ANOVAs. The patients were older and more likely to be female than controls. There was no significant difference in body mass index (BMI) between patients and controls (*p* = 0.91).

### 3.2. Comparing Patients and Healthy Controls

A higher sedimentation rate (SR) (*p* = 0.003), leukocytes (*p* < 0.001), lymphocytes (*p* < 0.001), neutrophils (*p* = 0.003), monocytes (*p* = 0.005), ferritin (*p* = 0.008), vitamin B12 (*p* < 0.001), P-calcium *p* = 0.005), alanine aminotransferase (ALAT) (*p* = 0.002), alkaline phosphatase (ALP) (*p* = 0.033), low-density lipoprotein cholesterol (LDL-cholesterol) (*p* = 0.001), and free T4 (thyroxine) (*p* < 0.001) were found among patients compared to controls. Lower potassium (*p* < 0.001), creatinine (*p* = 0.016), and CK (*p* < 0.001) were found in patients than in controls. The results are shown in Table 2.

### 3.3. Comparing Subgroups of Patients between Blood Tests, Clinical Characteristics, and Case Criteria for ME/CFS

Comparing blood results across function impairment status among patients revealed a significant difference in potassium levels (*p* = 0.048), CK (*p* < 0.001) and creatinine (*p* = 0.018), all variables increasing with decreasing function level while ASAT (*p* = 0.042) and ALAT (*p* = 0.023) decreased with more severely impaired function level. Longer illness duration was only significantly associated with potassium levels (*p* = 0.007) that decreased with longer duration. The only difference between different diagnostic criteria was a higher creatinine (*p* = 0.015) among the “non-ICC” group. The results are shown in Table 3.

## 4. Discussion

Our main findings include a lower creatin kinase (±CK) value in routine blood samples among patients than among controls, and lower in those with severe function impairment ME/CFS compared to moderate and mild function impairment. This is in line with the results by Nacul et al. [11]. No differences in blood results were found when comparing categories of illness duration, which could have potentially explained these CK result, because inactivity is known to cause muscle loss and therefore could potentially have influenced the CK level. CK is an enzyme important for energy production, especially in tissues with high and fluctuating energy demands, such as the brain, skeletal muscle, and heart. One of the functions is to maintain constant levels of ATP, acting as a transport mechanism [55]. Measures of CK in the blood might indicate the availability of cellular energy [56]. While an elevated CK is more thoroughly studied, a low CK has been reported less frequently [11], but might be associated with muscle weakness in rheumatoid arthritis [57]. In studies of Huntington’s disease, it has been suggested that the loss of CK in the brain may be an important factor for reduced brain energy [11]. As Nacul et al. have suggested, the low concentration in CK among ME/CFS patients, could reflect abnormalities in energy metabolism, which could explain the exertion intolerances that are often reported by patients. Alternatively, it could result from physical inactivity [11]. Our results may indicate that CK could be a possible candidate as a potential marker for ME/CFS. However, the level was within the reference range and there are many factors that can influence CK to be used as a biomarker. It should also be emphasized that CK measured in plasma represents CK from skeletal muscles. Thus, the role of CK in ME/CFS patients should be further explored in future studies.

The creatinine level was also significantly lower among patients than controls and related to severity of impairment with lower levels in those with more severe impairment. This is also found by Nacul et al. that suggested a possible explanation of a low creatinine being the result of poor conversion of creatinine phosphate to creatinine in muscle by CK that could explain the low levels of creatinine [11]. The creatinine level was also lower among the ICC group those fulfilling all case criteria including the most stringent ICC-criteria compared to the non-ICC group those fulfilling only Fukuda definition and CCC, similarly to what we see comparing patients to healthy controls. In the absence of a similar correlation for CK, this difference in diagnosis is harder to explain by such a mechanism.

We also found results somewhat in line with a potentially increased inflammation, with sedimentation rate, leukocytes, lymphocytes, monocytes, and ferritin being higher among patients. The difference, although being highly significant, is small throughout and within the normal range—i.e., 0.53-point difference for leukocytes—and might not be clinically relevant. These findings are similar to those of previous studies, e.g., Bates et al. [32] and may possibly support the idea of a low-grade inflammation in ME/CFS patients. Furthermore, other inflammation parameters, such as CRP, were not significantly different between groups.

Vitamin B12 was higher among patients than controls, but among patients there was no correlation with severity, duration or diagnostic criteria. A possible explanation for this may be that ME/CFS patients frequently take dietary supplements [19]. Unfortunately, we did not register intake of dietary supplements and/or concomitant medications and thus lack such data. It has been suggested that a B12 supplement could be beneficial for ME/CFS patients [58].

An unfavorable lipid profile has been described among ME/CFS patients [18]. We discovered an increase in low density lipoproteins-cholesterol (LDL-cholesterol). However, these values are still within normal ranges.

Other results that are more difficult to explain are moderately lower potassium among patients, which decreased with severity and illness duration, and increased calcium and protein. ALAT was also higher among patients, and both ALAT and ASAT increased with severity. The differences are small, however, and may not be clinically significant, although statistically so. No comparable research that could explain these differences with any certainty is known to the authors. Furthermore, we conducted a large number of analyses and thus some of our results may have occurred by random (we did not apply Bonferroni tests).

### Strengths and Limitations

The patients were older and more likely to be female than the controls, but this was controlled for in the statistical analyses. Female gender is more common in the ME/CFS population, and this is therefore representative for this population. There was also a difference in level of education between patients and controls, but education was not related to any of the dependent variables and thus not corrected for. We did not collect information about potential differences in muscle mass and physical activity from study participants. This could potentially be relevant for the observed difference in circulating CK and creatinine levels in study patients. A significant difference in BMI was not found. BMI is not an indicator of body composition.

We did not include patients younger than 18 or older than 65 years or those using another language than Norwegian. Generalization to recent immigration groups and other age groups, for example, should therefore be made with caution.

Furthermore, Aker Hospital is a tertiary care center to which patients with complex symptoms, co-morbid somatic or psychiatric conditions and patients who are difficult to manage in routine clinical contexts are mostly referred. Therefore, generalizing to ME/CFS patients as a group should be done with caution. The interdisciplinary diagnostic evaluation procedures were extensive however, so we regard it as most/extremely likely that our patients are correctly diagnosed, and this is a strength of this study. Other strengths are a relatively high number of ME/CFS patients and healthy donoors, and a large variety of variables.

Our participation rate is estimated to be around 40%, but we unfortunately lack data allowing us to compare participants with non-participants (e.g., with respect to demographic and clinical characteristics). We are aware that many listed patients are feeling too ill/fatigued to participate, and this could imply that we have a selection bias against those who are most severely affected. On the other hand, ME/CFS is characterized by symptom fluctuations and it may also have been the case that those who eventually could not participate as agreed in the data collection experienced a bad period with a transient symptom increase without necessarily having a permanently low level of function. In this regard, around 28% of the patients are in the group with the lowest level of function. This is about what we find it in the general ME/CFS population [8].

## 5. Conclusions

Results of several routine blood tests of ME/CFS patients differed from those healthy controls. Our findings particularly highlight that decreased creatinine and CK levels may indicate greater muscle damage and metabolic disturbances in ME/CFS patients and is worthy of future studies. This is also true of results that may indicate a possible low-grade inflammation in ME/CFS patients.

## Figures and Tables

**Table 1 jcm-10-03105-t001:** Mean ± standard deviation, and T-tests conducted on demographic and clinical characteristics for ME/CFS patients and healthy controls.

Variables	ME/CFS Patients (*n* = 149)	Healthy Controls (*n* = 264)	*p*-Values (Patients vs. Controls)
Age (years) 28 missing	37.7±11.4	31.1±8.4	<0.001
Gender,			
Male *n* (%)	28 (19)	78 (30)	0.02
Female *n* (%)	121 (81)	186 (70)	
Education (years completed), 59 missing			
1–10 years (%)	16 (12)	2 (1)	<0.001
10–14 years (%)	52 (35)	59 (29)	
14–16 (%)	59 (40)	87 (42)	
>16 (%)	21 (13)	58 (27)	
BMI (kg/m^2^) (26 missing)	24.5 (4.7)	24.5 (4.2)	0.91

**Table 2 jcm-10-03105-t002:** Mean scores for routine blood tests in patients with ME/CFS and healthy controls.

	ME/CFS Patients		Healthy Controls		Patients vs. Controls		
Variables	Mean ± SD (95% CI)	*n*	Mean ± SD (95% CI)	*n*	Mean Difference (95% CI)	Eta *	*p*-Values
Sedimentation rate (5.88/4.13)	6.28 ± 7.47 (5.558;7.0003)	123	4.90 ± 5.29 (4.274;5.525)	215	1.38 (0.045;2.714)	0.161	0.003
Hemoglobin (1.08/13.38)	13.85 ± 1.16 (13.704;13.997)	142	13.85 ± 1.02 (13.723;13.976)	228	0.001 (0.819;1.941)	0.001	0.99
Erythrocytes (4.50/0.42)	4.65 ± 0.43 (4.591;4.71)	142	4.66 ± 0.41 (4.605;4.707)	228	0.01 (1.291;1.469)	0.001	0.89
Hematocrit (0.40/0.03)	0.41 ± 2.81 (0.408;0.417)	141	0.41 ± 0.03 (0.41;0.418)	228	0 (−0.252;0.252)	0.031	0.50
MCHC (33.43/0.93)	33.65 ± 0.96 (33.48;33.81)	142	33.51 ± 0.91 (33.37;33.656)	228	0.14 (−1.765;2.045)	0.063	0.20
MCV (89.19/3.63)	88.85 ± 5.29 (88.219;89.481)	141	89.20 ± 4.72 (88.656;89.746)	227	0.35 (−1.799;2.499)	0.045	0.38
Thrombocytes (242.91/53.84)	241.82 ± 74.41 (232.16;251.487)	140	236.84 ± 50.99 (228.512;245.17)	228	4.98 (−9.153;19.113)	0.045	0.41
Leukocytes (5.42/1.46)	5.76 ± 1.74 (5.504;6.024)	139	5.17 ± 1.37 (4.946;5.393)	228	0.59 (0.248;0.932)	0.187	<0.001
Lymphocytes (1.91/0.59)	2.03 ± 0.65 (1.925;2.133)	142	1.76 ± 0.53 (1.672;1.852)	227	0.27 (0.145;0.395)	0.210	<0.001
Neutrophils (2.92/1.02)	3.11 ± 1.24 (2.928;3.292)	139	2.77 ± 1.06 (2.618;2.931)	226	0.34 (0.089;0.591)	0.105	0.003
Monocytes (0.42/0.13)	0.46 ± 0.16 (0.439;0.484)	140	0.42 ± 0.12 (0.401;0.4441)	226	0.39 (0.309;0.371)	0.148	0.005
Eosinophils (0.17/0.12)	0.18 ± 0.11 (0.154;0.199)	142	0.18 ± 0.11 (0.155;0.195)	228	0.04 (0.017;0.063)	0.001	0.912
Basophils (0.02/0.04)	0.03 ± 0.05 (0.023;0.038)	141	0.02 ± 0.13 (0.016;0.029)	227	0.002 (0.017;0.021)	0. 110	0.072
hsCRP (2.50/2.48)	2.7 ± 4.61 (2.093;3.319)	87	2.10 ± 4.88 (1.551;2.629)	129	0.61 (−0.668;1.888)	0.110	0.112
TIBC (63.51/10.27)	60.57 ± 6.84 (56.799;64.332)	38	63.50 ± 9.91 (60.601;66.392)	64	2.93 (−1.43;7.29)	0.138	0.171
Ferritin (70.59/56.67)	110.45 ± 80.81 (91.389;129.509)	37	82.32 ± 54.33 (68.036;96.609)	64	28.13 (−1.41;–57.67)	0.265	0.009
Vitamin B12 (351.17/135.23)	409.7 ± 222.61 (385.366;434.060)	135	326.24 ± 113.6 (305.516;346.954)	229	83.43 (62.89;124.21)	0.281	<0.001
Vitamin B9 (19.19/7.72)	18.36 ± 9.53 (16.909;19.816)	131	18.72 ± 7.22 (17.486;19.949)	226	0.36 (−1.544;–2.264)	0.001	0.693
Sodium (104.15/1.85)	140.34 ± 2.13 (140.013;140.667)	144	140.28 ± 1.68 (139.999;140.566)	230	0.06 (−0.16;–0.27)	0.001	0.779
Potassium (3.82/0.22)	3.81 ± 0.26 (3.77;3.845)	143	3.92 ± 0.25 (3.884;3.949)	229	0.11 (0.056;0.164)	0.235	<0.001
P-calcium (2.30/0.075)	2.3 ± 0.08 (2.307;2.33)	143	2.30 ± 0.25 (2.286;2.308)	230	0.02 (−0.02;0.06)	0.145	0.005
S-calcium (1.22/0.034)	1.22 ± 0.03 (1.217;1.229)	143	1.22 ± 0.08 (1.212;1.223)	228	0 (−0.02;–0.06)	0.077	0.138
Phosphate (1.03/0.17)	1.03 ± 0.17 (1.002;1.061)	144	1.01 ± 0.17 (0.984;1.035)	230	0.02 (0;0.038)	0.020	0.240
ASAT (21.63/5.68)	22.03 ± 15.3 (21.044;23.008)	142	22.79 ± 8.64 (21.944;23.643)	227	0.76 (−2.035;3.545)	0. 063	0.211
ALAT (19.32/9.77)	23.24 ± 26.06 (21.511;24.960)	142	19.94 ± 13.35 (19.444;21.426)	228	3.3 (−1.37;7.971)	0.158	0.002
CK (94.06/66.91)	84.33 ± 34.43 (73.708;95.061)	144	121.90 ± 282.88 (112.559;131.231)	224	37.57 (0.282;75.422)	0.281	<0.001
Creatinine (68.46/11.03)	70.16 ± 11.59 (68.435;71.886)	143	72.76 ± 11.88 (71.263;74.259)	227	2.6 (0.126;5.074)	0.126	0. 016
HDL-cholesterol (1.54/0.38)	1.37 ± 0.4 (1.227;1.502)	38	1.50 ± 0.4 (1.395;1.608)	63	0.13 (−0.303;0.043)	0.176	0.082
LDL-cholesterol (2.77/0.42)	3.16 ± 0.88 (2.921;3.393)	37	2.71 ± 0.7 (2.530;2.894)	63	0.45 (0.104;0.796)	0.324	0.001
Triglycerides (1.03/0.53)	1.29 ± 1.44 (1.096;1.474)	36	1.10 ± 0.43 (0.953;1.247)	63	0.19 (−0.312;0.692)	0.176	0.084
Albumin (43.38/2.56)	43.91 ± 2.95 (43.055;44.757)	38	44.23 ± 2.25 (43.577;44.885)	64	0.32 (2.154;2.795)	0.071	0.500
Total protein (70.80/3.51)	72.53 ± 3.6 (71.277;73.784)	38	71.00 ± 8.57 (70.032;71.962)	63	1.53 (−0.945;4.005)	0.214	0.033
TSH (1.92/0.94)	1.93 ± 0.98 (1.756;2.101)	143	1.89 ± 1.34 (1.739;2.040)	220	0.04 (−0.081;0.279)	0.001	0.721
IgG4 (0.49/0.45)	0.64 ± 0.38 (0.471;0.809)	36	0.54 ± 0.29 (0.413;0.669)	62	0.1 (−0.082;0.282)	0.110	0.296
IgA (2.03/0.77)	2.22 ± 0.82 (1.929;2.515)	37	2.13 ± 0.96 (1.903;2.348)	62	0.1 (−0.182;0.475)	0.063	0.559
IgM (1.08/0.41)	1.06 ± 0.62 (0.904;1.218)	35	0.96 ± 0.47 (0.841;1.074)	62	0.1 (−0.142;0.342)	0.122	0.241
Rheumatoid factor IgA (2.55/2.68)	2.89 ± 12.99 (2.412;3.368)	133	2.61 ± 0.96 (2.205;3.017)	221	0.28 (−2.021;2.305)	0.055	0.348
Rheumatoid factor IgM (5.51/4.74)	6.02 ± 13.87 (5.155;6.883)	137	5.26 ± 4.73 (4.521;5.996)	227	0.76 (−1.743;3.263)	0.077	0.156
Anti-CCP (0.85/0.77)	0.94 ± 0.48 (0.641;1.239)	37	0.88 ± 0.6 (0.655;1.104)	63	0.06 (−0.159;0.279)	0.032	0.717

* Values are presented as means ± standard deviation (SD) including confidence interval (95% CI) controlled for gender and age differences by ANCOVA. Effect sizes (Eta) by OLS.

**Table 3 jcm-10-03105-t003:** Comparison of baseline mean scores for routine blood tests among ME/CFS patients based on functional status assessment, illness duration, and case criteria for ME/CFS among participants.

	Function Status				Illness Duration				Case Criteria		
Variables (Mean ± SD)	Severe (*n* = 43)	Moderate (*n* = 75)	Mild (*n* = 31)	Beta Eta	1–2 Years (*n* = 14)	>2 Years (*n* = 118)	Since Childhood (*n* = 17)	Beta Eta	Non-ICC (*n* = 55)	ICC (*n* = 94)	Beta
Sedimentation rate (7.24 ± 4.46)	6.09	5.88	6.38	0.017 0.045 (ns)	6.70	6.30	3.92	−0.129 0.184 (ns)	5.86	6.16	0.039 (ns)
Hemoglobin (13.3 ± 1.17)	13.85	13.89	14.00	0.043 0.055 (ns)	13.58	13.91	14.09	0.085 0.13 (ns)	13.90	13.91	−0.002 (ns)
Erythrocytes (4.46 ± 0.43)	4.65	4.63	4.70	0.042 0.084 (ns)	4.56	4.65	4.67	0.037 0.077 (ns)	4.65	4.64	−0.011 (ns)
Hematocrit (0.40 ± 0.034)	0.41	0.41	0.42	0.059 0.089 (ns)	0.40	0.41	0.42	0.076 0.015 (ns)	0.41	0.41	0.029 (ns)
MCHC (33.45 ± 0.96)	33.65	33.77	33.54	−0.035 0.10 (ns)	33.81	33.67	33.76	0.009 0.055 (ns)	33.80	33.61	−0.088 (ns)
MCV (89.47 ± 3.41)	88.90	89.09	89.40	0.050 0.055 (ns)	88.44	89.11	89.58	0.078 0.077 (ns)	88.74	89.41	0.098 (ns)
Thrombocytes (242.45 ± 58.26)	248.99	244.11	238.64	−0.057 0.063 (ns)	230.13	245.27	247.14	0.048 0.077 (ns)	238.38	248.60	0.08 (ns)
Leukocytes (5.87 ± 1.55)	5.78	5.75	5.52	−0.055 0.063 (ns)	5.27	5.79	5.56	0.022 0.10 (ns)	5.62	5.78	0.043 (ns)
Lymphocytes (2.05 ± 0.66)	2.09	1.98	1.99	−0.058 0.071 (ns)	1.72	2.05	1.98	0.081 0.14 (ns)	1.99	2.02	0.027 (ns)
Neutrophils (3.20 ± 1.11)	3.02	3.15	2.87	−0.034 0.10 (ns)	2.89	3.11	2.86	−0.035 0.089 (ns)	3.02	3.09	0.023 (ns)
Monocytes (0.45 ± 0.15)	0.47	0.45	0.44	−0.052 0.063 (ns)	0.41	0.46	0.45	0.042 0.089 (ns)	0.43	0.47	0.14 (ns)
Eosinophils (0.17 ± 0.11)	0.18	0.18	0.18	−0.021 0.032 (ns)	0.21	0.17	0.20	−0.024 0.122 (ns)	0.19	0.17	−0.10 (ns)
Basophils (0.029 ± 0.045)	0.034	0.032	0.029	−0.041 0.044 (ns)	0.028	0.033	0.031	0.004 0.032 (ns)	0.028	0.035	0.071 (ns)
hsCRP (1.39 ± 1.86)	0.82	1.15	1.62	0.15 0.15 (ns)	1.05	1.27	0.67	−0.045 0.11 (ns)	1.33	1.032	−0.076 (ns)
Vitamin-B12 (407.74 ± 163)	440.31	386.16	415.44	−0.058 0.14 (ns)	428.75	405.09	382.37	−0.074 0.063 (ns)	406.79	402.31	−0.021 (ns)
Vitamin-B9 (19.21 ± 8.89)	20.72	16.94	18.55	−0.10 0.18 (ns)	19.23	18.53	15.54	−0.12 0.12 (ns)	18.89	7.64	−0.070 (ns)
Sodium (140.03 ± 2.13)	104.20	140.70	139.95	−0.025 0.15 (ns)	140.99	140.49	139.63	−0.13 0.15 (ns)	140.66	140.23	−0.093 (ns)
Potassium (3.77 ± 0.24)	3.76	3.82	3.87	0.16 * 0.17	4.01	3.80	3.77	−0.21 ** 0.27 **	3.82	3.81	−0.019 (ns)
P-calcium (2.30 ± 0.075)	2.33	2.32	2.32	−0.087 0.11 (ns)	2.35	2.32	2.30	−0.14 0.16 (ns)	2.33	2.31	−0.12 (ns)
S-calcium (1.22 ± 0.034)	1.23	1.22	1.22	−0.060 0.063 (ns)	1.23	1.22	1.23	0.046 0.11 (ns)	1.22	1.23	0.076 (ns)
Phosphate (1.03 ± 0.17)	1.042	1.022	1.021	−0.046 0.055 (ns)	1.078	1.019	1.034	−0.024 0.095 (ns)	1.019	1.033	0.048 (ns)
ASAT (20.92 ± 5.24)	23.092	22.13	20.51	−0.17 * 0.17	22.11	22.24	20.67	−0.068 0.095 (ns)	21.61	22.31	0.065 (ns)
ALAT(21.063 ± 11.67)	26.83	23.031	20.58	−0.19 * 0.20	21.93	24.17	20.26	−0.067 0.12 (ns)	22.53	25.097	0.056 (ns)
CK (67.66 ± 34.50)	64.85	87.091	88.34	0.25 *** 0.34 ***	68.41	82.55	88.032	0.11 0.15 (ns)	87.32	77.75	−0.14 (ns)
Creatinine (65.79 ± 10.75)	68.089	70.81	73.38	0.17 * 0.2	65.53	71.34	71.14	0.097 * 0.18	72.90	68.99	−0.18 *
TSH (1.94 ± 0.98)	1.82	1.99	2.16	0.12 0.12 (ns)	1.66	2.040	1.87	0.041 0.12 (ns)	2.034	1.94	−0.043 (ns)
Rheumatoid factor-IgA (2.76 ± 2.64)	2.55	2.95	2.41	−0.0090 0.089 (ns)	2.94	2.58	3.63	0.063 0.12 (ns)	3.24	2.42	−0.15 (ns)
Rheumatoid factor-IgM (6.12 ± 4.56)	6.12	5.47	6.62	0.028 0.1 (ns)	5.98	5.79	6.095	0.016 0 (ns)	6.47	5.42	−0.11 (ns)

Means and *t*-tests controlled for age, gender, and BMI among the participants. Values are presented as mean ± SD. A significance threshold was set at * *p* < 0.05, ** *p*< 0.01, and *** *p* < 0.001.

## Data Availability

Data and material are available from the corresponding author.

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
