# Peer review of "Evaluating Routine Blood Tests According to Clinical Symptoms and Diagnostic Criteria in Individuals with Myalgic Encephalomyelitis/Chronic Fatigue Syndrome"

_jcm, 2021, doi:10.3390/jcm10143105_

Round 1

Reviewer 1 Report

Dear Authors,

first of all, I thank You for giving me the opportunity to read this Your manuscript, submitted for publication. 

I read it with interest.

Here are my comments and suggestions. I hope that they are useful for You. 

MAIOR  COMMENTS:

My major comments are two:

1) You did not assess any of the possible co-morbidities. This could have introduced an inclusion bias, distorcing some of the results You reported;

2) Data You reported had no practical significance, and the statistical significance and significativity was also very modest.       

MINOR   COMMENTS  AND  SUGGESTIONS:

1)  Some typographical errors should be corrected. For instance, in lines 41,82,129,141,145, 146 (and so on.....), spaces must be corrected. The typographical character of Most (line 49) should be reduced;  

2) In Introduction section, estimated annual cost for Danish and Spanish patients should be reported;

3) Measures should be better clarified. A summary table can be very useful for the readers; 

4) Assessment of laboratory inflammatory markers was difficult to understand.  Indeed, C-reactive protein concentrations were not significantly different between groups. Please, clarify this point;

5)  What You wrote in lines 49-50  seemed contradicted by what You wrote in lines 50-62. Please, clarify this point. 

Author Response

Responses to the comments of Reviewer 1:

Comments:

1) You did not assess any of the possible co-morbidities. This could have introduced an inclusion bias, distorting some of the results You reported. 

Response: We are grateful for this comment. Excluding patients with possible co-morbidities, i.e. somatic and/or psychiatric conditions that could explain their symptoms, in practice may have introduced a “bias” of sorts, but hedged against reporting results that were not related to ME/CFS.

2) Data You reported had no practical significance, and the statistical significance and significativity was also very modest. 

Response: Thank you for the possibility to comment on the significance of the results. The practical significance of our results is very hard to assess, but some may indeed be relevant to diagnostics, treatment or care, depending on one’s theoretical orientation and/or professional background. With our relatively low N and modest statistical power even results that are barely statistically significant may indeed direct our attention to associations with profound theoretical implications.   

3)  Some typographical errors should be corrected. For instance, in lines 41,82,129,141,145, 146 (and so on.....), spaces must be corrected. The typographical character of Most (line 49) should be reduced.

Responses: Thank you for pointing this out. We have corrected the typos- throughout the manuscript.

4) In Introduction section, estimated annual cost for Danish and Spanish patients should be reported.

Response: Thank you for pointing this out. We agree with this comment. Therefore, we have added a few sentences about the economic burden in Europe. There are no Europe-wide prevalence data on ME/CFS, but a commonly held belief is that there are some 250,000 sufferers in the UK (7). If this is correct then there may be some 2.5 million ME/CFS patients in Europe as a whole. It often results in diminished functionality and increased economic impact. The economic impact of an illness is generally divided into two categories: direct and indirect costs. Despite high prevalence rates and the disabling nature of the illness, few if any studies have examined the economic impact of ME/CFS at the individual and societal costs across Europe. The following sentences have been added in text at page 2, introduction part, first paragraph in the revised version of the manuscript:

“ME/CFS is estimated to affect over 2.5 million of adults across Europe. It often results in diminished functionality and increased economic impact. Despite high prevalence rates and the disabling nature of the illness, few studies have examined the economic impact of ME/CFS at the individual and societal costs across Europe”.

The reference to this statement is a recently published paper that is added as reference 9 in the revised version of the manuscript and in the reference list. Pheby et al. The development of a consistent Europe-wide approach to investigating the economic impact of ME/CFS: a report from the EUROMENE. Healthcare 2020; 8:88. DOI: 10.3390/healthcare8020088.

3) Measures should be better clarified. A summary table can be very useful for the readers.

 Response: Thank you for making us aware of this. We agree that the measures are somewhat vaguely described and have now tried to clarify that we only use DSQ as a measure (questionnaire) in the current study. Thus we have re-written the text in section 2.3 (page 4, paragraph 1, line 209) with the following sentence: “Data were collected from the self-report questionnaire DSQ and blood tests. Questions from the DSQ were applied for categorization of patients groups entered in the statistical analyses.”

4) Assessment of laboratory inflammatory markers was difficult to understand.  Indeed, C-reactive protein concentrations were not significantly different between groups. Please, clarify this point

Response: Thank you for this comment. It is correct that CRP concentrations were not significantly different between any of the groups. This is commented upon in the discussion section, page 12, paragraph three, lines 460 to 461) where the following sentence have been added at the end of this paragraph in the revised version of the manuscript: “Furthermore, other inflammation parameters, such as CRP, were not significantly different between groups.”

5)  What You wrote in lines 49-50 seemed contradicted by what You wrote in lines 50-62. Please, clarify this point. 

Response: Thank you for this comment. These are simply references to previous research that tend to show inconsistent or contradictory results.

Reviewer 2 Report

The authors provide important data regarding laboratory values in a large, well characterized cohort of patients with ME/CFS. The paper is well written. 

There are only a few points to correct (comma is lacking between glucose and phosphate  line 57, Table format is shifted for some columns, especially table 1). 

Author Response

Review 2:

There are only a few points to correct (comma is lacking between glucose and phosphate line 57, Table format is shifted for some columns, especially table 1): 

Response: Thank you for this comment. These errors have been changed in the text and in the tables in the revised version of the manuscript.

Reviewer 3 Report

The above study is highly needed in ME/CFS field as a try to reveal biomarker for ME/CFS. Moreover, the aim it to examine differences between groups in relation to criteria used in diagnosis, what should also be explored due to the heterogeneity of symptoms profile and severity in ME/CFS patients. Authors have done a nice job on the manuscript.

Overall, one major limitation of the study may be reflected by lack of application of control for false discovery rate (FDR) or FWER of results described, as multiple hypothesis are tested. Rest of my comments/suggestion are minor.

In overall, paper in the current form need editorial corrections some of them I have pointed out below.

Abstract line 26 put comma „aspartate, ALAT”

„ More studies are needed to explore this.” Please rephrase this sentence to make it more specific or delete it”

Introduction „post exertional malaise” please write as „post-exertional malaise”

Lines 49-66 please extend all of the acronyms used here.

Line 122 „CME/CFS” is that” a typo?

„2.4. Groups” I would suggest to change the title of this paragraph. Title could be misleading, because paragraph has no description of healthy controls, while it is included into analysis. Moreover, I would suggest to change names of group 1 and group 2 to ICC and non-ICC, respectively. In addition, I suppose that maybe it would be necessary to extend this part. What is the difference in symptoms profile in ICC vs non-ICC?

2.7 Statistical analysis

From “As a general rule, t-tests require” to ending of this paragraph I lost my tracking. Are You using t-test here? Dependent or independent, because the characteristics of assumption on normality is different between those? “yielded almost identical p-values” using which statistical tests? Please provide name of the first vs name of the second. Could You cite some literature that would justify such approach?

I suppose that You should apply method to control for false discovery rate (FDR) or FWER?

Results:

“therefore corrected for in ANOVAs.” ANOVAs or ANCOVAs?

“There was no significant difference in body mass index 214 (BMI) between patients and controls (P=0.31).” P-value in Table 1 states otherwise? I suppose that p-value should be denoted by „p” or „p” not „P”.

I would suggest to not bold out anything in tables. You have described that p-value <0.05 are bolded, but it seems to be otherwise. You should add description of effect size in paragraph 2.7. Table 2 consist of column “Variable (Mean/S/E) „ what it shows, where mean and SE comes in this column? Why there are no units of measurements provided?”. Do You think that some figures could be applied to illustrate between group differences?

Discussion:

“Other results that are more difficult to explain are moderately lower potassium 290 among patients, which decreased with severity and illness duration, an increased calcium 291 and protein.” Could You extend this paragraph? What might be the pathological mechanism underlying those observations?

Study limitations

“A significant difference in BMI was not found.” I would add that BMI is not an indicator of body composition

Conclusions:

“Several routine blood tests of ME/CFS „ I would add „Results of several routine blood tests of ME/CFS“. Moreover, I would extend this part to answer all hypothesis that were indicated in the last paragraph of Introduction section.

Author Response

Review 3

1) Abstract line 26 put comma „aspartate, ALAT”.

Response: This has been corrected in the text

2) “More studies are needed to explore this.” Please rephrase this sentence to make it more specific or delete it”

Response: Thank you for the comment. This sentence has been deleted. 

Introduction

1) „post exertional malaise” please write as „post-exertional malaise”.

Lines 49-66 please extend all of the acronyms used here.

Line 122 „CME/CFS” is that” a typo?

Response: Thank you for these comments. Changes have been made accordingly. 

 2) „2.4. Groups” I would suggest to change the title of this paragraph. Title could be misleading, because paragraph has no description of healthy controls, while it is included into analysis. Moreover, I would suggest to change names of group 1 and group 2 to ICC and non-ICC, respectively. In addition, I suppose that maybe it would be necessary to extend this part. What is the difference in symptoms profile in ICC vs non-ICC? 

Response: Thank you for these suggestions with which we agree. The title of the paragraph has been changed from “Groups” to “Patient groupings: functional status, illness duration and diagnostic criteria” and the names of the groups have been changed to “ICC” and “non-ICC” in the revised version of the manuscript both in text (page 4, paragraph, lines 211 and lines 232 to 235) as welll as in the headings of table 2.

2.7 Statistical analysis

1) From “As a general rule, t-tests require” to ending of this paragraph I lost my tracking. Are You using t-test here? Dependent or independent, because the characteristics of assumption on normality is different between those? “yielded almost identical p-values” using which statistical tests? Please provide name of the first vs name of the second. Could You cite some literature that would justify such approach?

Response: We are grateful for this comment because this section could be somewhat misleading in its present form. We do indeed use t-tests in Tables 2 and 3 (ANOVA, ANCOVA, OLS) using blood samples as dependent variables. Of course, complications arising from only dependent variables that are not normally distributed are rather modest while outliers / extreme scores can be very problematic.                                                             We have changed the text in the section starting with  “As a general rule, t-tests require “ to: “Mean differences between groups in continuous variables when tested by t-tests (as in the present paper) typically require normally distributed variables within groups, but there is considerable robustness to deviations from normality overall. Log-transformed versions of dependents were visually inspected and yielded almost identical p-values (for t-tests) and within-group means (also when adjusted by ANCOVAs) are reported using the untransformed metric (page 5, paragraph two, lines 284 to 289) in the revised version of the manuscript..

2) I suppose that You should apply method to control for false discovery rate (FDR) or FWER? 

Response: Thank you for this question. With the very high number of dependents and relatively modest Ns FDRs would leave us with virtually nothing to report. We should regard our study as exploratory rather than as one testing hypotheses.

Results:

3) “therefore corrected for in ANOVAs.” ANOVAs or ANCOVAs?

Response: Thank you for this comment. We have used both ANOVA and ANCOVA. Please see response 1, paragraph 2,7 above.

4) “There was no significant difference in body mass index 214 (BMI) between patients and controls (P=0.31).” P-value in Table 1 states otherwise? I suppose that p-value should be denoted by „p” or „p” not „P”. 

Response: Thank you for pointing this out. Uppercase p has been changed to lowercase in the p-value in the text and in all tables. The p-value in the text has been changed in accordance with the correct BMI p-value in table 1 from 0.31 to 0.91.

5) I would suggest to not bold out anything in tables. You have described that p-value <0.05 are bolded, but it seems to be otherwise.

Response: We agree with the reviewer’s comment. The sentences about “bold in text” are now deleted in the table text.

6) You should add description of effect size in paragraph 2.7. Table 2 consist of column “Variable (Mean/S/E) „ what it shows, where mean and SE comes in this column? Why there are no units of measurements provided?”. Do You think that some figures could be applied to illustrate between group differences?

Response: We thank the reviewer for this comment. Units of measurement are provided by means and standard deviations in the first column of Table 2 and Table 3 (there was a misprint in Table 2 which has been corrected to “Means/SD”). In accordance with the reviewer’s comment, the following sentence has been added to paragraph 2.7 (page 5, paragraph, lines 290-291).: Units of measurement for the dependent variables are shown as means and standard deviations in Table 2 and 3.

Effect sizes that are standardized, i.e. comparable across dependents, have been utilized. For dichotomous independents (e.g. “patients vs. controls”) this is done by “eta”, i.e. the square root of the “eta squared” routinely reported in ANOVAs and ANCOVAs and corresponding to “variance explained”. Effect sizes for tests of linear trends are reported by standardized betas (OLS).

The following sentence has been added to the end of paragraph 2.7 (page 5, lines 292 to 296): “Effect sizes for dichotomous and trichotomous independents in Table 2 and 3 are cited as “eta”, i.e. the square of root of the variance explained by the groups comprising an independent, controlling for possible covariates. Effect sizes for linear trends in ordered trichotomous independents (function status, illness duration), are cited as standardized betas obtained by OLS.

Illustrating between-group differences by figures probably would not add much given our use of standardized effect sizes, but presenting z-scores for subgroups for the 4 or 5 most dramatic results might be interesting.

Discussion:

1.“Other results that are more difficult to explain are moderately lower potassium 290 among patients, which decreased with severity and illness duration, an increased calcium 291 and protein.” Could You extend this paragraph? What might be the pathological mechanism underlying those observations? 

Response: Thank you for this comment. We state in the paragraph that we do not know of any studies that could explain these results. We have discussed this in our research group and unfortunately we still do not know of any empirical studies that may explain such mechanisms.

Study limitations

“A significant difference in BMI was not found.” I would add that BMI is not an indicator of body composition.

Response: We agree with the comment of the reviewer. The suggested sentence has now been added to the text in the revised version of the manuscript (page 12, line 412).  

Conclusions:

“Several routine blood tests of ME/CFS „ I would add „Results of several routine blood tests of ME/CFS“. Moreover, I would extend this part to answer all hypothesis that were indicated in the last paragraph of Introduction section. 

Response: Thank you for this comment. The proposed sentence has been inserted in the text in accordance with the reviewer’s comment. As we regard this as an exploratory study we did not write any hypotheses in the introduction section that we addressed in the conclusion. Hopefully this is acceptable.

Round 2

Reviewer 1 Report

Dear Authors, 

You chose to write speculative and generic phrases instead to responding to my comments. 

I am sorry for this. 

Author Response

Revised responses to the comments of Reviewer 1:

Comments:

1) You did not assess any of the possible co-morbidities. This could have introduced an inclusion bias, distorting some of the results You reported. 

Response: We are grateful for this comment. In accordance with the diagnostic criteria for ME/CFS the included patients did not (and shall not) suffer of any co-morbid psychiatric or somatic disorders. Thus patients with comorbidities were excluded in the evaluation and inclusion process in order to fulfill the ME/CFS criteria that were applied in the present study. Nevertheless, in principle in a population study based on self-report we might have included and assessed ME/CFs patients both with and without other somatic/psychiatric disorders. However, in the present study we wanted to make sure that the ME/CFS symptoms were not caused by or due to other somatic/psychiatric disorders.

2) Data You reported had no practical significance, and the statistical significance and significativity was also very modest. 

Response: Thank you for the possibility to comment on the significance of the results. The practical significance of studies like ours is often hard to assess. Nevertheless, our findings could be helpful to clinicians who may have to consider what type of blood tests are worthwhile should be performing and assessing. Furthermore, the nature of our results possibly may inspire and generate hypotheses and future studies of potential pathological mechanisms that could allow for a better understanding ME/CFS, even preparing the ground for novel clinical trials. We would also like to emphasize that research studies like ours have been requested by other significant researchers in the field (Jason, personal communication). With our relatively low N and modest statistical power even results that are barely statistically significant may indeed direct our attention toward associations with profound theoretical implications. 

3)  Some typographical errors should be corrected. For instance, in lines 41,82,129,141,145, 146 (and so on.....), spaces must be corrected. The typographical character of Most (line 49) should be reduced.

Responses: Thank you for pointing this out. We have corrected the typos- throughout the manuscript.

4) In Introduction section, estimated annual cost for Danish and Spanish patients should be reported.

Response: Thank you for pointing this out. We agree with this comment. However, we are unaware of any studies reporting the said annual costs. Nevertheless, we have now commented upon this crucial point by adding a few sentences about the economic burden in Europe.

There are no Europe-wide prevalence data on ME/CFS, but a commonly held belief is that there are some 250,000 sufferers in the UK (7). If this is correct then there may be some 2.5 million ME/CFS patients in Europe as a whole. ME/CFS often results in diminished functionality and with considerable economic impact. Despite high prevalence rates and the disabling nature of the illness, few if any studies have examined the economic impact of ME/CFS in Europe. Particularly we are not aware of any studies reporting estimated annual cost for Danish and Spanish patients. The following sentences have been added in text at page 2, introduction part, first paragraph in the revised version of the manuscript:

“ME/CFS is estimated to affect over 2.5 million of adults across Europe. The condition often results in diminished functionality and increased economic impact. Despite high prevalence rates and the disabling nature of the illness, few studies have examined the economic impact of ME/CFS at the individual level and the societal costs across Europe”.

The reference to this statement is a recently published paper that has been added as reference 9 in the revised version of the manuscript and in the reference list. (Pheby et al. The development of a consistent Europe-wide approach to investigating the economic impact of ME/CFS: a report from the EUROMENE. Healthcare 2020; 8:88. DOI: 10.3390/healthcare8020088.)

Hopefully our response to your comment is sufficient.

3) Measures should be better clarified. A summary table can be very useful for the readers.

 Response: Thank you for making us aware of this. We agree that the measures are somewhat vaguely described and have now tried to clarify that we only use DSQ as a measure (questionnaire) in the current study. Thus we have re-written the text in section 2.3 (page 4, paragraph 1, line 209) with the following sentence: “Data were collected from the self-report questionnaire DSQ and blood tests. Questions from the DSQ were applied for categorization of patients groups that were included in the statistical analyses.”

4) Assessment of laboratory inflammatory markers was difficult to understand.  Indeed, C-reactive protein concentrations were not significantly different between groups. Please, clarify this point

Response: Thank you for this comment. It is correct that CRP concentrations were not significantly different between any of the groups. This is commented upon in the discussion section, page 12, paragraph three, lines 460 to 461) where the following sentence have been added at the end of this paragraph in the revised version of the manuscript: “Furthermore, other inflammation parameters, such as CRP, were not significantly different between groups.”

5)  What You wrote in lines 49-50 seemed contradicted by what You wrote in lines 50-62. Please, clarify this point. 

Response: Thank you for this comment. Our summary in the introduction simply refers to previous research and studies that overall tend to show inconsistent or contradictory results.

Round 3

Reviewer 1 Report

Dear Authors,

please, read my previous comments and suggestions.

Author Response

Review 1, response 3:

This manuscript cannot be published with the abstract in the current form. The abstract needs to be clear and reflect the findings of the study.

Thank you for the comments on the abstract:

  1. Add that the healthy controls were recruited from blood donors.

Response: This has now been added to the abstract (line 18).

  1. Do not say "patients were subgroupd in accoradance with various case definitions, funciton impairmene tstatus and illness duration" and do not explain what they are. If you don't have space in the abstract to talk about the subgroups results, then just omit this from the abstract. They are considered secondary analyses

Response:The subgroups are removed from the abstract

  1. Need to give the size of the differences (effect size maybe?) and the significance of the changes (95% CI). Cannot just say "patients had higher ........" or "patients had lower.....".

Response: CIs have been added to Table 2 in order to illustrate the nature of mean differences between patients and controls. Eta (with corresponding p-value) is used as effect parameter. The following is placed in text at page 6: “For illustrative purposes confidence intervals (CI) have been added after controlled means for patients and controls in Table 2.” (in line 214/215) and the following sentence: “Levels of significance for effect sizes are cited as exact p-values (Table 2) and routinely categorized (Table 3)” (in line 220/221). Borg p-values and CI are placed in the result part of the abstract.

  1. Please submit a clean copy. The copy I read had all tracked changes and it is very difficult to read.

Response: We have now submitted a clean copy (with all the changes accepted in the manuscript) and a second version with the latest revisions.
